# Efficient Self-Evaluation for Diffusion Language Models via Sequence Regeneration

## Abstract

Diffusion large language models (dLLMs) have recently attracted significant attention for their ability to enhance diversity, controllability, and parallelism. However, their non-sequential, bidirectionally masked generation makes quality assessment difficult, underscoring the need for effective self-evaluation. In this work, we propose DiSE, a simple yet effective self-evaluation confidence quantification method for dLLMs. DiSE quantifies confidence by computing the probability of regenerating the tokens in the entire generated sequence, given the full context. This method enables more efficient and reliable quality assessment by leveraging token regeneration probabilities, facilitating both likelihood estimation and robust uncertainty quantification. Building upon DiSE, we further introduce a flexible-length generation framework, which adaptively controls the sequence length based on the model's self-assessment of its own output. Experiments demonstrate that DiSE consistently improves performance across multiple datasets, increasing likelihood evaluation by $4.0\%$ and uncertainty evaluation by $6.4\%$, while achieving up to a $32\times$ speedup over Monte Carlo simulation baseline, and additionally improving flexible-length generation accuracy. These results establish DiSE as an efficient and versatile self-evaluation framework for diffusion-based language models.

## 1 Introduction

Recently, diffusion large language models (dLLMs) (Yu et al., 2025) have emerged as a promising direction in natural language processing. In contrast to auto-regressive (AR) models, dLLMs adopt the generative framework of diffusion models (Ho et al., 2020; Nichol & Dhariwal, 2021; Song et al., 2020), framing text generation as a progressive denoising process. This approach enables better diversity, controllability, and parallel generation compared to AR models. Nonetheless, the non-sequential and bidirectional nature of dLLMs makes direct likelihood-based self-evaluation challenging (Nie et al., 2025). Concurrently, self-evaluation has been recognized as a fundamental capability of LLMs, serving as the basis for a wide range of applications such as hallucination detection (Shorinwa et al., 2025; Fadeeva et al., 2024), answer quality assessment (Chang et al., 2024), and generation quality enhancement (Huang et al., 2024; Xie et al., 2024).

In AR models, causal masking enforces a strict left-to-right generation order, allowing sequence probability to be decomposed into token-level conditional probabilities. This simplifies the generation process and enables self-evaluation through likelihood estimation. In contrast, dLLMs use bidirectional masking and a non-sequential, stepwise generation process, making direct likelihood-based self-evaluation challenging. Currently, dLLMs rely primarily on Monte Carlo simulation-based approximations of sequence likelihood (Nie et al., 2025), but this method is computationally expensive and often yields suboptimal estimates, limiting its practical effectiveness. Moreover, owing to the intrinsic token-level self-evaluation signal provided by next-token prediction in AR models, the generation length can be adaptively controlled via real-time EOS token prediction. Unlike AR models, conventional dLLMs lack such an effective built-in likelihood-based self-evaluation signal, which forces them into fixed-length generation and fundamentally restricts their flexibility.

In this work, we propose DiSE, a simple yet effective self-evaluation confidence quantification method for diffusion large language models. DiSE is derived by feeding the entire sequence back into the dLLM and computing the probability of regenerating its tokens under the full context. This

method enables the model to assess its own generation quality by evaluating how well it can reproduce the original sequence when conditioned on the entire context, effectively leveraging its own internal predictions. Based on DiSE, we introduce a flexible-length sequence generation method that, unlike conventional fixed-length generation, enables controllable and adaptive output lengths guided by the model's self-assessment. Serving as a real-time self-evaluation mechanism, DiSE guides the process of searching, assessing and stopping to determine the optimal generation length.

DiSE provides a versatile mechanism for dLLMs, acting as an effective estimator for conditional likelihood evaluation and facilitating robust uncertainty quantification (Shorinwa et al., 2025). This approach significantly improves computational efficiency while achieving higher evaluation accuracy compared to traditional Monte Carlo simulation-based methods. Extensive experiments on likelihood evaluation, uncertainty quantification, and flexible-length generation show the effectiveness of the proposed DiSE.

Our main contributions are summarized as follows:

- **Efficient and Reliable Self-evaluation Mechanism for dLLMs.** We propose DiSE, a simple yet effective self-evaluation confidence quantification method for diffusion large language models, which enables dLLMs to perform efficient and reliable self-assessment by computing the probability of sequence regeneration.
- **Flexible-length dLLM Generation with DiSE.** We propose a flexible-length generation framework for dLLMs based on DiSE, which enables adaptive sequence lengths through real-time self-evaluation and is validated through extensive experiments.
- **Performance Improvements in Likelihood Evaluation and Uncertainty Quantification.** The DiSE consistently enhances dLLM performance by serving as an efficient estimator for conditional likelihood evaluation and improving uncertainty quantification. It achieves a $4.0\%$ improvement in average accuracy on ARC-Challenge and GPQA, and a $6.4\%$ improvement in average ROC-AUC across Countdown, GSM8K, MATH500 and SVAMP, while yielding a $32\times$ speedup over Monte Carlo simulation.

## 2 RELATED WORK

### 2.1 DLLMS

Diffusion Large Language Models (dLLMs) (Yu et al., 2025) adapt the diffusion modeling paradigm (Ho et al., 2020; Nichol & Dhariwal, 2021; Song et al., 2020), which is originally successful in image and video generation (Podell et al., 2023; Zhong et al., 2025), to natural language. Early efforts, such as D3PM (Austin et al., 2021), DiffusionBERT (Austin et al., 2021), RDM (Zheng et al., 2023), MDLM (Sahoo et al., 2024) and MD4 (Shi et al., 2024), focused on exploring training objectives, noise scheduling strategies, and parameterization methods. Recent research includes LLaDA (Nie et al., 2025), the first large-scale dLLM, DIFFUSION-LLMs (Ye et al., 2023) with multi-stage training strategies, and DiffuGPT / DiffuLLaMA (Gong et al., 2024), which adapt pre-trained auto-regressive models to the diffusion framework. DREAM (Ye et al., 2025) further demonstrates strong performance in complex reasoning tasks. Subsequent developments, such as LLaDA 1.5 (Zhu et al., 2025) with variance-reduced preference optimization for preference alignment and TESS 2 (Tae et al., 2025) with auto-regressive initialization and adaptive noise scheduling, further improve generation quality.

### 2.2 SELF-EVALUATION FOR LLMS

Self-evaluation (Ren et al., 2023; Geng et al., 2023) has emerged as a crucial mechanism in LLMs, providing models with the capability to assess the reliability of their own outputs and to produce internal measures of confidence and correctness. Self-evaluation is most directly performed via likelihood estimation, using the model's probabilistic output to quantify plausibility. While sequence likelihood is a natural evaluation signal for AR models, it is generally intractable for dLLMs. Recent efforts (Nie et al., 2025) address this by developing approximate likelihood measures, but their effectiveness is often limited by computational cost and estimation variance. Beyond likelihoods, uncertainty quantification (UQ) (Shorinwa et al., 2025; He et al., 2023; Vashurin et al., 2024) evaluates the confidence of model predictions and plays a key role in mitigating hallucinations in risk-aware

settings. Token-level approaches estimate uncertainty from the conditional probability distribution of the generated tokens, employing entropy-based metrics, sequence normalization, or meaning-aware scoring (e.g., perplexity (Shorinwa et al., 2025), CCP (Fadeeva et al., 2024), MARS (Bakman et al., 2024)) for more fine-grained assessments. Self-verbalized UQ (Stengel-Eskin et al., 2024; Xu et al., 2024; Lin et al., 2022) encourages LLMs to articulate their confidence through verbalized probabilities or epistemic markers. Building on these uncertainty signals, recent work leverages self-evaluation for calibration, aligning model confidence with empirical accuracy and thereby improving the reliability and quality of generated outputs (Huang et al., 2024; Xie et al., 2024).

## 3 PRELIMINARIES

### 3.1 AUTO-REGRESSIVE LLM PROBABILITY ESTIMATION

Given an auto-regressive language model and a text sequence $X = (x_1, x_2, \ldots, x_N)$, the probability of generating the entire sequence is factorized as the product of conditional probabilities:

$$p_\theta(X) = \prod_{i=1}^{N} p_\theta(x_i \mid x_{<i}), \tag{1}$$

where $x_{<i} = (x_1, \ldots, x_{i-1})$ represents all preceding tokens, and $\theta$ denotes the model parameters. This factorization allows exact computation of the sequence probability by multiplying the model's predicted probabilities for each token given its context. The probability estimation for conditional generation is detailed in Appendix B.1.

### 3.2 DLLM MONTE CARLO PROBABILITY ESTIMATION

DLLMs do not employ the causal masking used in auto-regressive LLMs and therefore the probability of generating a sequence cannot be factorized as a simple product of conditional probabilities. To approximate the log-probability of generating a target sequence $X^0 = (x_1^0, x_2^0, \ldots, x_N^0)$, the traditional approach (Nie et al., 2025) adopts the following term:

$$\mathbb{E}_{l, X^l} \left[ \frac{N}{l} \sum_{i=1}^{N} \mathbf{1} \left[ x_i^l = \langle \text{mask token} \rangle \right] \log p_\theta \left( x_i^0 \mid X^l \right) \right], \tag{2}$$

where $l$ is uniformly sampled from $\{1, 2, \ldots, N\}$, and $X^l = (x_1^l, x_2^l, \ldots, x_N^l)$ is obtained by uniformly sampling $l$ tokens from $X^0$, replacing the tokens at these positions with mask tokens, while keeping all other tokens identical to those in $X^0$. Since the exact computation of this expectation is intractable, Monte Carlo simulation (Harrison, 2010) is employed, where a finite number of samples are generated and the expectation is approximated by their empirical average. This approximation enables tractable estimation of sequence probabilities for dLLMs. The probability estimation for conditional generation is detailed in Appendix B.2.

## 4 METHOD

### 4.1 DISE

In traditional likelihood estimation approaches, whether using auto-regressive LLMs or dLLMs with Monte Carlo simulation, the common paradigm is to condition on the tokens at known positions and predict the tokens at unknown positions based on their probability distributions. However, under the dLLM framework, it is also possible to predict the tokens at positions that are already known. In this work, we propose DiSE, a self-evaluation confidence quantification method for dLLMs that employs token regeneration probability as a novel indicator of model confidence and investigate different token sets to calculate token regeneration probability.

Let the text sequence be $X = (x_1, x_2, \ldots, x_N)$. The dLLM takes $X$ as input and concurrently predicts the tokens at all positions that already exist. $p_\theta(x_i \mid X)$ represents the probability of the model regenerating token $x_i$ at position $i$ given the entire sequence $X$. Accordingly, the probability of the model regenerating $X$ given $X$ is formulated as $\prod_{i=1}^{N} p_\theta(x_i \mid X)$. Consider a binary mask

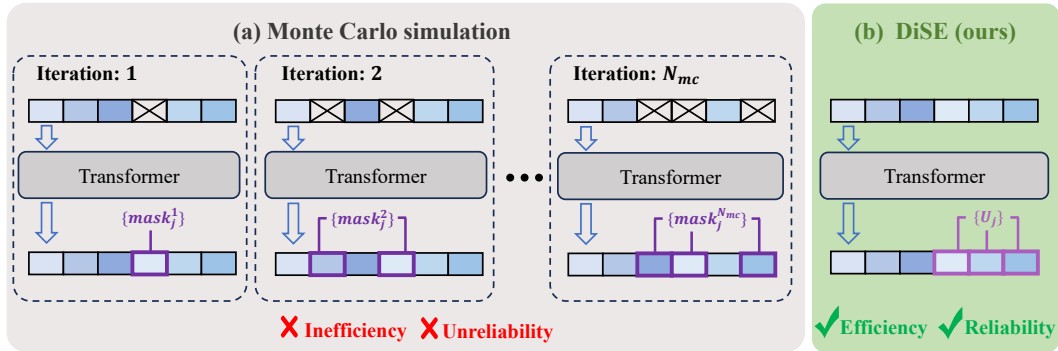

Figure 1: A simplified illustration of self-evaluation confidence quantification methods for clarity. (a) Monte Carlo simulation approach for dLLMs. A total of $N_{mc}$ simulations are performed. In the $i$-th simulation, a set of masked positions $\{mask_j^i\}$ is sampled. The tokens at these positions are replaced with mask tokens, and the model predicts the probability of correctly generating these tokens. The final estimation is obtained by aggregating the results across all $N_{mc}$ simulations. (b) The proposed DiSE for dLLMs. The set of selected positions $\{U_j\}$ is predefined. The model receives the entire sequence and estimates the regeneration probability of the tokens at $\{U_j\}$.

$M \in \{0,1\}^N$, where $M_i = 1$ indicates that the token at position $i$ is included in the probability calculation for regeneration, and $M_i = 0$ means it is ignored. Let $U = \{i \mid M_i = 1\}$ be the index set of the selected positions. The probability of regenerating the tokens in the selected region is formulated as $\prod_{i \in U} p_\theta(x_i \mid X)$. After taking the logarithm and averaging over the number of selected tokens, the DiSE score is defined as follows:

$$\text{DiSE}(X) = \frac{1}{|U|} \sum_{i \in U} \log p_\theta(x_i \mid X), \tag{3}$$

where different selection modes are employed to determine the binary mask $M \in \{0,1\}^N$, thereby controlling the index set of selected positions $U$. This measure captures the model's confidence in regenerating its own tokens and allows flexible evaluation over either local regions or the entire sequence. For conditional generation with prompt $P$ and generated response $R$, the DiSE score is calculated by treating the concatenated sequence $[P; R]$ as $X$. Figure 1 presents a simplified visualization of the Monte Carlo simulation approach for dLLMs and the proposed DiSE.

### 4.2 OBSERVATION

**Observation I: Semantic Coherence Positively Correlates with DiSE Scores.** We sample 15 well-formed sentences and generate fully randomized versions by replacing all original tokens with random tokens. The DiSE scores are computed for both the natural and randomized sentences using a binary mask $M$ with all positions set to one, corresponding to the selection mode 'full'. As shown in Figure 2 (a), natural sentences achieve substantially higher DiSE scores than their randomized counterparts. Additionally, we perform three local token randomization experiments, replacing 10 tokens in the front, middle or back regions of each sentence, and the DiSE scores are measured for these perturbed positions. In these experiments, the selection modes are denoted as 'first-10'/'mid-10'/'last-10', indicating that $M = 1$ is applied only to the respective region. Figures 2 (b), (c) and (d) show that natural sentences consistently obtain higher DiSE scores than randomized sentences in all regions. These findings indicate that DiSE effectively captures semantic coherence of both global and local region, allowing fine-grained self-evaluation across different parts of a sentence.

**Observation II: Answer Accuracy Positively Correlates with DiSE Scores.** We conduct a series of experiments on four commonly used reasoning datasets: Countdown (Pan et al., 2025), GSM8K (Cobbe et al., 2021), MATH500 (Lightman et al., 2023) and SVAMP (Patel et al., 2021). The model outputs are categorized into two groups according to whether the generated answers match the ground-truth solutions. We compute the DiSE scores separately for the correct and incorrect groups and report their averages under two selection modes 'full' and 'last-10'. The results, summarized in Figure 3, consistently reveal that correct outputs tend to exhibit higher DiSE scores

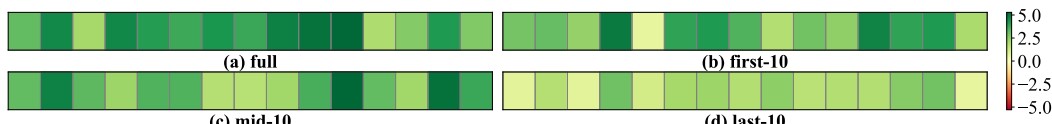

Figure 2: Differences between the DiSE scores of natural sentences and randomized sentences using the LLaDA-Instruct-8B model under four selection modes: 'full' (entire sentence), 'first-10' (first 10 tokens), 'mid-10' (10 tokens from the middle) and 'last-10' (last 10 tokens). Each subfigure contains 15 blocks, representing 15 sampled sentences. All blocks are shown in green (difference > 0), indicating that natural sentences consistently achieve higher DiSE scores than randomized sentences.

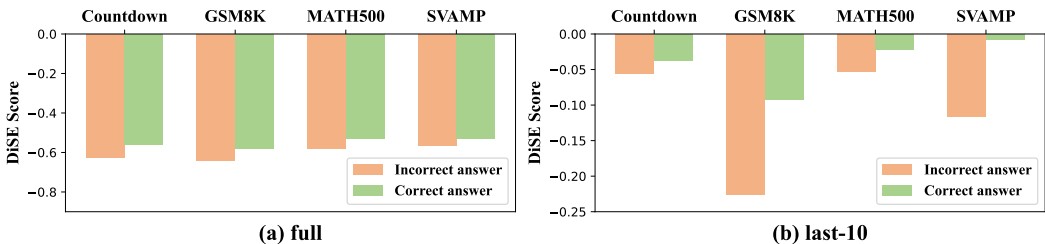

Figure 3: Comparison between the DiSE scores of correct and incorrect answers across four datasets using the LLaDA-Instruct-8B model under two selection modes 'full' and 'last-10'. Correct outputs consistently show higher values, with the disparity notably amplified under 'last-10', which focuses on the final ten tokens associated with answer positions.

than incorrect ones across different datasets. Importantly, under the selection mode 'last-10', which focuses on the final ten tokens closely associated with the answer positions, the disparity between correct and incorrect outputs is substantially amplified. This finding highlights the strong correlation between DiSE scores and answer accuracy, supporting the reliability of the proposed DiSE.

### 4.3 DIRECT USE OF DISE

**Conditional Likelihood Estimation for dLLMs.** Conditional likelihood estimation serves as an important metric for evaluating the generative ability of language models. During the evaluation, we estimate the probability or log-probability of generating a candidate response $R$ conditioned on a given prompt $P$. For each prompt $P$, there may be multiple candidate responses, and we select the one with the highest probability as the final answer and compute the accuracy accordingly. In this work, DiSE is employed as an approximate estimator of the conditional likelihood evaluation via the unconventional regenerating probability, rather than the standard generating probability.

**Uncertainty Quantification for dLLMs.** Quantifying the uncertainty of model outputs is crucial for assessing their reliability. In the context of dLLMs, we use the DiSE score as a self-evaluation signal to measure the confidence of a generated sequence. Sequences with higher DiSE scores are considered more reliable, while lower scores indicate higher uncertainty. The negative of the DiSE score is used to quantify the uncertainty of the model output, with a higher value reflecting a higher estimated uncertainty.

### 4.4 FLEXIBLE-LENGTH DLLM GENERATION WITH DISE

In general, dLLMs require the generation length $L$ to be fixed and specified in advance. Different choices of $L$ lead to different outcomes, and longer generations incur higher computational costs. In our work, we aim to relax the restriction of a fixed generation length and instead allow the output length to be adjusted flexibly within a controllable range. This is enabled by DiSE, which provides an intrinsic signal to evaluate the quality of generations without ground-truth supervision. Leveraging this property, we propose flexible-length dLLM generation.

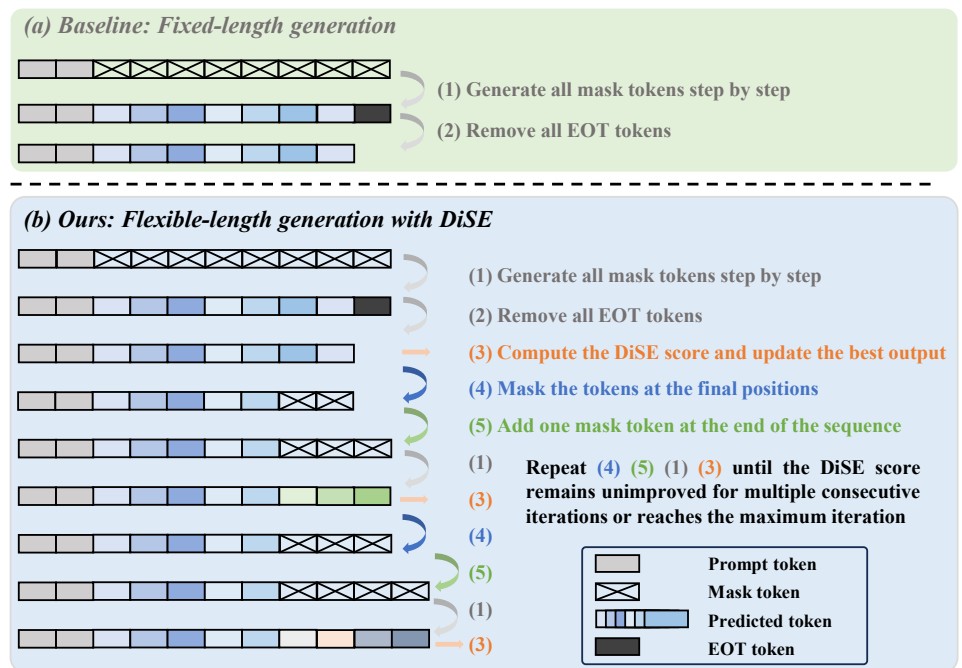

Figure 4: Illustration of the flexible-length dLLM generation framework with DiSE versus fixed-length generation. (a) Fixed-length generation baseline. The model generates a sequence of pre-determined length $L$. (b) Flexible-length generation with DiSE. Starting from base length $L$, DiSE serves as a real-time self-evaluation mechanism, assessing the quality of the current sequence and deciding whether to terminate the extension process.

Our method proceeds as follows. Given a prompt $P$ and a base length $L$, we first generate an initial response $R$ of length $L$. Let $\overline{R}$ denote the sequence obtained by removing all EOT tokens from $R$. We construct the complete token sequence as $X^{(1)} = [P; \overline{R}]$ and compute its DiSE score, which serves as the guiding criterion for controlling the generation length. Keeping the tokens in the early positions unchanged, we apply a masking operation to the last $D$ tokens, and add one additional mask token at the end of the sequence. We use the model to regenerate the sequence, after which the DiSE score of the newly generated sequence is computed. At each iteration, $D$ is incremented by one. This process is repeated iteratively, with DiSE determining whether the extended generation is beneficial. If the DiSE score improves, we retain the extension; otherwise, if the DiSE score remains unimproved for $K$ consecutive iterations, we stop. To avoid unbounded computation, we set a maximum of $M_{max}$ iterations. The overall procedure is illustrated in Figure 4 and the detailed algorithm is provided in Appendix C. This flexible-length generation process uses the DiSE score as a self-evaluation signal, enabling dLLMs to adaptively decide their output length in a principled manner.

## 5 EXPERIMENTS

### 5.1 EXPERIMENTAL SETUP

Experiments are conducted using two dLLMs, LLaDA-Instruct-8B (Nie et al., 2025) and LLaDA-1.5-8B (Zhu et al., 2025), on a diverse set of datasets, including ARC-Challenge (Clark et al., 2018), GPQA (Rein et al., 2024), Countdown (Pan et al., 2025), GSM8K (Cobbe et al., 2021), MATH500 (Lightman et al., 2023) and SVAMP (Patel et al., 2021). The conventional Monte Carlo simulation approach for dLLMs is used as the baseline, with the number of samples $N_{mc}$ evaluated under two settings: $N_{mc} = 1$ and $N_{mc} = 32$. Additionally, we include the auto-regressive LLM LLaMA3-Instruct-8B (Dubey et al., 2024) for comparison in the experiments. More details are presented in Appendix D.

Table 1: Conditional likelihood estimation results on ARC-Challenge and GPQA. The table compares the proposed DiSE against the Monte Carlo simulation baseline with varying $N_{mc}$, and also includes a comparison with the probability estimates from auto-regressive LLMs. The last column reports the average number of model forward passes per computation.

| | Method | ARC-Challenge | GPQA | # forward passes |
|---|---|---|---|---|
| | MC, $N_{mc} = 1$ | 0.306 | 0.212 | 1 |
| LLaDA-Instruct-8B | MC, $N_{mc} = 32$ | 0.478 | 0.286 | 32 |
| | **DiSE (ours)** | **0.542** | **0.301** | 1 |
| | MC, $N_{mc} = 1$ | 0.311 | 0.203 | 1 |
| LLaDA-1.5-8B | MC, $N_{mc} = 32$ | 0.488 | 0.275 | 32 |
| | **DiSE (ours)** | **0.567** | **0.299** | 1 |
| LLaMA-3-8B | probability | 0.530 | 0.304 | 1 |

Table 2: ROC-AUC scores for uncertainty quantification on the Countdown, GSM8K, MATH500 and SVAMP datasets with varing generation lengths. The table compares Monte Carlo simulation baseline with varying $N_{mc}$, the proposed DiSE, and the perplexity calculation using the auto-regressive model LLaMA3-Instruct-8B. The last column reports the average ROC-AUC scores across the preceding 12 settings.

| | | Countdown | | | GSM8K | | | MATH500 | | | SVAMP | | | Avg. ROC-AUC↑ |
|---|---|---|---|---|---|---|---|---|---|---|---|---|---|---|
| | Method / Gen Len | 128 | 256 | 512 | 128 | 256 | 512 | 128 | 256 | 512 | 128 | 256 | 512 | |
| | MC, $N_{mc} = 1$ | 0.524 | 0.520 | 0.528 | 0.539 | 0.513 | 0.540 | 0.497 | 0.541 | 0.532 | 0.563 | 0.575 | 0.509 | 0.532 |
| LLaDA-Instruct-8B | MC, $N_{mc} = 32$ | **0.595** | **0.534** | 0.558 | 0.590 | 0.552 | 0.595 | 0.528 | 0.578 | 0.531 | 0.616 | 0.551 | 0.647 | 0.573 |
| | **DiSE (ours)** | 0.578 | 0.521 | **0.622** | **0.633** | **0.644** | **0.658** | **0.611** | **0.634** | **0.604** | **0.688** | **0.692** | **0.755** | **0.637** |
| | LLaMA perplexity | 0.574 | 0.419 | 0.392 | 0.675 | 0.605 | 0.577 | 0.575 | 0.637 | 0.551 | 0.686 | 0.650 | 0.590 | 0.578 |
| | MC, $N_{mc} = 1$ | 0.525 | **0.588** | 0.528 | 0.516 | 0.559 | 0.525 | 0.558 | 0.514 | 0.525 | 0.562 | 0.466 | 0.554 | 0.535 |
| LLaDA-1.5-8B | MC, $N_{mc} = 32$ | 0.608 | 0.557 | 0.520 | 0.559 | 0.578 | 0.608 | 0.580 | 0.546 | **0.551** | 0.585 | 0.513 | 0.597 | 0.567 |
| | **DiSE (ours)** | **0.610** | 0.471 | **0.586** | **0.610** | **0.616** | **0.613** | **0.606** | **0.553** | 0.533 | **0.599** | **0.629** | **0.677** | **0.592** |
| | LLaMA perplexity | 0.596 | 0.459 | 0.362 | 0.635 | 0.631 | 0.546 | 0.652 | 0.588 | 0.550 | 0.639 | 0.587 | 0.620 | 0.572 |

## 5.2 Conditional Likelihood Estimation

We evaluate the performance of our proposed approach in the conditional likelihood estimation experiments, with the results summarized in Table 1. Compared to the conventional Monte Carlo simulation baseline, our method demonstrates substantial and consistent improvements on ARC-Challenge and GPQA, indicating its reliability as a estimator in likelihood evaluation. Moreover, when contrasted with the probability estimates obtained from auto-regressive LLMs, the proposed approach achieves comparable or even superior results. We also report the average number of model forward passes required for each computation. Notably, relative to the Monte Carlo baseline with $N_{mc} = 32$, our method achieves nearly a 32-fold increase in computational efficiency while demonstrating enhanced predictive performance. Specifically, using the LLaDA-Instruct-8B model, our method outperforms the Monte Carlo baseline with $N_{mc} = 1$, which offers comparable efficiency, by 23.6% on ARC-Challenge and 8.9% on GPQA. Furthermore, even when compared to the higher-cost Monte Carlo baseline with $N_{mc} = 32$, our approach achieves nearly a 32× speedup while still improving performance, with gains of 6.4% on ARC-Challenge and 1.5% on GPQA.

## 5.3 Uncertainty Quantification

For uncertainty quantification experiments, we evaluate the ability to distinguish correctness among multiple generated answers for each question using ROC-AUC scores (Kuhn et al., 2023), where the ROC-AUC score measures the probability that a randomly chosen correct answers receives lower uncertainty than a randomly chosen incorrect one. We generate 5 answers per question. Table 2 presents the results on the Countdown, GSM8K, MATH500 and SVAMP datasets with varying generation lengths, using ROC-AUC scores to assess uncertainty quantification performance. In comparison to the conventional Monte Carlo simulation method, our approach yields substantial improvements. Using the LLaDA-Instruct-8B model, our method improves average ROC-AUC by 10.5% across twelve generation settings over the Monte Carlo method with $N_{mc} = 1$ at comparable cost. Even compared to Monte Carlo with $N_{mc} = 32$, which incurs a nearly 32× higher cost, our approach remains superior by 6.4%. Compared to the perplexity method of an auto-regressive LLM,

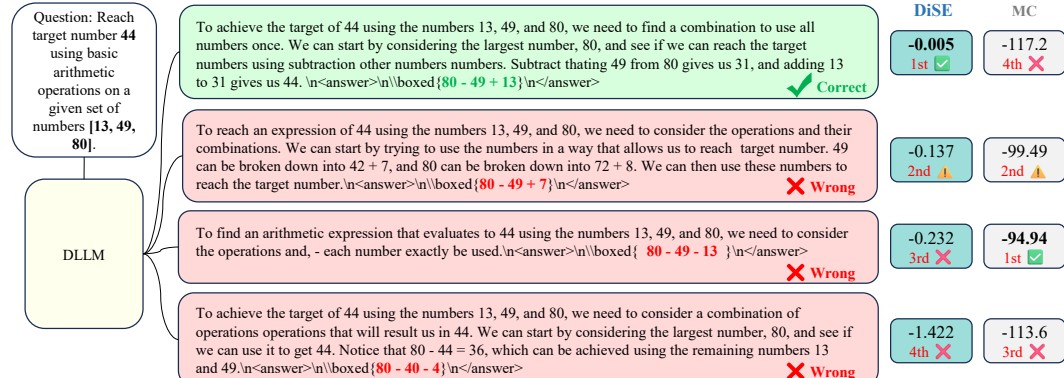

Figure 5: Qualitative example of uncertainty quantification with four generated answers using LLaDA-Instruct-8B. DiSE assigns higher score to the correct answer and lower scores to incorrect answers, while the Monte Carlo simulation ($N_{mc} = 32$) produces scores that do not consistently reflect correctness.

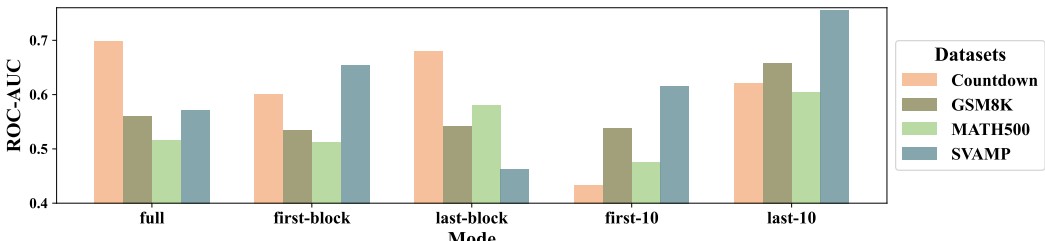

Figure 6: Ablation study of different DiSE selection modes for uncertainty quantification using LLaDA-Instruct-8B with a generation length of 512. Each mode corresponds to a different subset of tokens used for regeneration probability calculation across the sequence. Results across multiple datasets show that focusing on the last few non-EOT tokens yields higher ROC-AUC scores for uncertainty quantification.

our approach yields a $5.9\%$ gain on the same generations. Additional results of best-of-N sampling experiments for uncertainty quantification are presented in Appendix E.1.

We present a qualitative example in Figure 5 illustrating the contrast between DiSE and Monte Carlo simulation in capturing answer correctness. Four candidate answers generated by LLaDA-Instruct-8B using the same input are shown. DiSE consistently assigns lower scores to incorrect answers, corresponding to higher uncertainty, while Monte Carlo simulation with $N_{mc} = 32$ fails to reflect answer correctness. This case study highlights the reliability of DiSE as a fine-grained sequence-level uncertainty measure. Additional qualitative examples are presented in Appendix E.2.

We investigate the effect of different DiSE selection modes on uncertainty quantification, where each mode specifies the subset of tokens used for computing regeneration probability: 'full' (all tokens), 'first-block' (tokens in the first generation block), 'last-block' (tokens in the last generation block, including EOT tokens), 'first-10' (first 10 generated tokens) and 'last-10' (last 10 non-EOT tokens). As shown in Figure 6, focusing on the last 10 non-EOT tokens tends to yield higher ROC-AUC scores on multiple datasets, as these tokens typically correspond to the answer region. Regeneration probabilities of earlier tokens provide limited information about answer correctness, and including EOT tokens in the last generation block negatively impacts the uncertainty estimation. In general, this ablation study demonstrates that carefully selecting the token subset for the DiSE computation significantly affects the quality of uncertainty quantification. More details on the results under different selection modes are presented in Appendix E.3.

## 5.4 FLEXIBLE-LENGTH dLLM GENERATION

Table 3 presents the evaluation results of flexible-length dLLM generation on the Countdown, GSM8K, MATH500 and SVAMP datasets with multiple base lengths $L$. Two fixed-length baselines,

Table 3: Results of flexible-length dLLM generation with DiSE on the Countdown, GSM8K, MATH500, and SVAMP datasets with varying base lengths. The table presents two fixed-length baselines. The first, **Baseline**, generates sequences with the base length $L$, while the second, **Baseline (Max Len)**, generates sequences with the base length $L$ increased by the maximum number of iterations $M_{max}$. These are compared with the proposed flexible-length generation with DiSE (DiSE-flexible). The final column reports the average accuracy across the preceding 12 configurations.

| | | Countdown | | | GSM8K | | | MATH500 | | | SVAMP | | | Avg. Accuracy |
|---|---|---|---|---|---|---|---|---|---|---|---|---|---|---|
| | Method / Base Len | 128 | 256 | 512 | 128 | 256 | 512 | 128 | 256 | 512 | 128 | 256 | 512 | |
| LLaDA-Instruct-8B | Baseline | 26.17 | 15.23 | 12.50 | 68.01 | 76.65 | 79.23 | **26.20** | 32.80 | **36.80** | 84.67 | 85.00 | 83.67 | 52.24 |
| | Baseline (Max Len) | 25.00 | 16.41 | **15.62** | 69.29 | 76.80 | 78.85 | 25.60 | 31.60 | 36.40 | 85.33 | 84.67 | 83.00 | 52.38 |
| | **DiSE-flexible (ours)** | **27.73** | **18.36** | **15.62** | **70.96** | **79.68** | **79.30** | 26.00 | **33.60** | 36.60 | **87.33** | **86.00** | **84.33** | **53.79** |
| LLaDA-1.5-8B | Baseline | 24.22 | 15.62 | 17.19 | 70.51 | 77.48 | 79.53 | 26.80 | 34.00 | 36.80 | **87.00** | 84.67 | 86.67 | 53.37 |
| | Baseline (Max Len) | 24.22 | 17.58 | 17.58 | 71.95 | 78.77 | 79.53 | 25.80 | 34.20 | 37.00 | 86.33 | 83.00 | 86.33 | 53.52 |
| | **DiSE-flexible (ours)** | **26.17** | **19.53** | **22.27** | **72.33** | **79.53** | **80.06** | **27.20** | **35.60** | **37.40** | **87.00** | **85.00** | **87.00** | **54.92** |

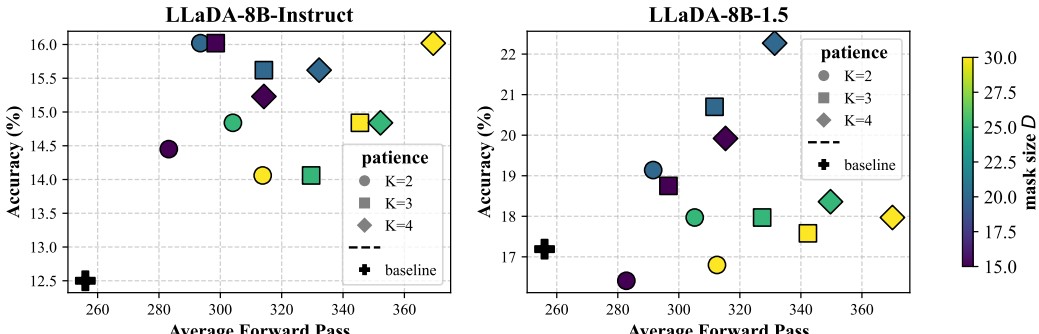

Figure 7: Ablation study on the Countdown dataset with base length $L = 512$ for flexible-length dLLM generation with DiSE, examining the effects of different patience $K$ and mask sizes $D$ on performance. The figure presents both accuracy and the average number of model forward passes for each configuration.

generating sequences of length $L$ or $L + M_{max}$, are considered to reflect conventional fixed-length generation. In contrast, our proposed method employs DiSE to guide flexible-length generation, enabling adaptive adjustment of the output sequence length. The results indicate that the flexible-length approach with DiSE yields average improvements over fixed-length baselines across multiple datasets and varing base lengths, providing strong evidence for the effectiveness of dynamically adapting sequence length with DiSE in dLLM generation.

To assess the impact of patience $K$ and mask size $D$, we perform an ablation on Countdown with base length $L = 512$, reporting accuracy and average forward passes in Figure 7. Our method generally outperforms the baseline, while different $K$ and $D$ settings highlight a trade-off between computational cost and performance. Additional ablation results are presented in Appendix F.

## 6 CONCLUSION

We introduce DiSE, a simple yet effective self-evaluation confidence quantification method for dLLMs. By employing token regeneration probability, DiSE achieves both high reliability and computational efficiency. Building upon DiSE, we propose a flexible-length generation framework, which enables adaptive sequence lengths through real-time self-evaluation. Comprehensive experiments across multiple datasets demonstrate the effectiveness of DiSE and the flexible-length generation framework with DiSE. DiSE closes the gap in dLLMs by introducing an efficient self-evaluation mechanism previously exclusive to auto-regressive LLMs. By leveraging DiSE, we overcome the fixed-length generation constraint in dLLMs and open the door to broader applications.

ETHICS STATEMENT

This work does not involve human subjects, personal data, or sensitive information. All datasets used in our experiments (ARC-Challenge, GPQA, Countdown, GSM8K, MATH500 and SVAMP) are publicly available benchmark datasets. We strictly adhered to ethical research practices and did not conduct any data collection that could raise privacy, security, or fairness concerns. Our work focuses on providing an efficient and reliable self-evaluation confidence quantification method for dLLMs, and introducing a flexible-length dLLM generation framework based on it, without introducing risks of harmful applications. To the best of our knowledge, this research complies with the ICLR Code of Ethics and poses no foreseeable ethical concerns.

REPRODUCIBILITY STATEMENT

We have made extensive efforts to ensure the reproducibility of our work. Comprehensive implementation details are reported in Section 5.1 and Appendix D. The detailed algorithm of flexible-length dLLMs generation with DiSE is provided in Appendix C. Upon acceptance, we will release the code of our method to facilitate replication and further research.

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

APPENDIX

# A LLM USAGE

In this section, we clarify the role of large language models (LLMs) in preparing this work. The model was used exclusively for language polishing, such as refining grammar, style, and readability, without contributing to the research design, analysis, or conclusions.

# B PROBABILITY ESTIMATION FOR CONDITIONAL GENERATION

## B.1 AUTO-REGRESSIVE LLM PROBABILITY ESTIMATION

In the context of conditional generation given a prompt $P$, let $R = (r_1, r_2, \ldots, r_N)$ denote the generated response of length $N$. The probability of generating $R$ given $P$ for an auto-regressive language model can be written as:

$$p_\theta(R \mid P) = \prod_{i=1}^{N} p_\theta(r_i \mid P, r_{<i}), \tag{S1}$$

where $r_{<i} = (r_1, \ldots, r_{i-1})$. This formulation allows exact computation of the probability of a model-generated response conditioned on a given prompt.

## B.2 DLLM MONTE CARLO PROBABILITY ESTIMATION

For dLLMs, let $R^0 = (r_1^0, r_2^0, \ldots, r_N^0)$ denote the generated response of length $N$. The traditional dLLM approach approximates the log-probability of generating $R^0$ given $P$ with the following term:

$$\mathbb{E}_{l,R^l} \left[ \frac{N}{l} \sum_{i=1}^{N} \mathbf{1} \left[ r_i^l = \langle \text{mask token} \rangle \right] \log p_\theta \left( r_i^0 \mid P, R^l \right) \right], \tag{S2}$$

where $l$ is uniformly sampled from $\{1, 2, \ldots, N\}$, and $R^l = (r_1^l, r_2^l, \ldots, r_N^l)$ is obtained by uniformly sampling $l$ tokens from $R^0$, replacing the tokens at these positions with mask tokens, while keeping all other tokens identical to those in $X^0$. Since the exact computation of this expectation is intractable, we employ Monte Carlo simulation to approximate it by sampling a finite number of instances and taking their empirical average.

# C ALGORITHM FOR FLEXIBLE-LENGTH DLLM GENERATION WITH DISE

We provide a detailed algorithm for the flexible-length dLLM generation framework guided by the DiSE score in Algorithm S1, which uses the DiSE score as a self-evaluation signal to achieve controllable sequence lengths and improved generation quality.

# D MORE IMPLEMENTATION DETAILS.

The datasets employed in our experiments are categorized into two groups: those used for conditional likelihood estimation and those intended for conditional generation. Specifically, we consider ARC-Challenge (Clark et al., 2018) and GPQA (Rein et al., 2024) for conditional likelihood estimation, which are challenging multiple-choice science question datasets. ARC-Challenge focuses on grade-school level questions that require advanced reasoning beyond simple retrieval, while GPQA contains expert-crafted questions in biology, physics, and chemistry that are difficult even for highly skilled humans and state-of-the-art AI models. For conditional generation, we use Countdown (Pan et al., 2025), GSM8K (Cobbe et al., 2021), MATH500 (Lightman et al., 2023), and SVAMP (Patel et al., 2021), which involve arithmetic and mathematical problems requiring step-by-step reasoning, advanced problem-solving, combinatorial thinking, and generalization across diverse problem formats. Regarding the selection mode, i.e., the binary mask $M$, we adopt different configurations for different datasets. For ARC-Challenge, we set $M = 1$ for the last two tokens of the prompt

---

**Algorithm S1** Flexible-length dLLM Generation with DiSE

---

**Require:** Prompt $P$, base length $L$, maximum iterations $M_{max}$, patience $K$, mask size $D$

**Ensure:** Final generated sequence $\hat{X}$

1: Generate an initial response $R$ of length $L$ given prompt $P$
2: Remove all EOT tokens from $R$ to obtain $\overline{R}$
3: $X^{(1)} = [P; \overline{R}]$
4: Compute initial confidence $s^{(1)} \leftarrow \text{DiSE}(X^{(1)})$
5: Set $\hat{X} \leftarrow X^{(1)}, \hat{s} \leftarrow s^{(1)}, t \leftarrow 1, c \leftarrow 0$
6: **while** $t < M_{max}$ **do**
7:    $t \leftarrow t + 1$
8:    Mask the last $D$ tokens of $X^{(t-1)}$ to obtain $X_m^{(t-1)}$
9:    Regenerate sequence $X^{(t)}$ from the masked input $[X_m^{(t-1)}; \langle \text{mask token} \rangle]$
10:    $s^{(t)} \leftarrow \text{DiSE}(X^{(t)})$
11:    **if** $s^{(t)} > \hat{s}$ **then**
12:       $\hat{X} \leftarrow X^{(t)}, \hat{s} \leftarrow s^{(t)}, c \leftarrow 0$
13:    **else**
14:       $c \leftarrow c + 1$
15:    **end if**
16:    **if** $c \geq K$ **then**
17:       **break**
18:    **end if**
19:    $D \leftarrow D + 1$
20: **end while**
21: **return** $\hat{X}$

---

Table S1: Best-of-N sampling results for uncertainty quantification on the Countdown, GSM8K, MATH500, and SVAMP datasets with varying generation lengths. The table compares the baseline without best-of-N sampling, Monte Carlo simulation with varying $N_{mc}$, the proposed DiSE, and the perplexity calculation using the auto-regressive model LLaMA3-Instruct-8B. The last column reports the average accuracy across the preceding 12 settings.

| | | Countdown | | | GSM8K | | | MATH500 | | | SVAMP | | | Avg. Accuracy |
|---|---|---|---|---|---|---|---|---|---|---|---|---|---|---|
| | Method / Gen Len | 128 | 256 | 512 | 128 | 256 | 512 | 128 | 256 | 512 | 128 | 256 | 512 | |
| | Baseline | 26.17 | 15.23 | 12.50 | 68.01 | 76.65 | 79.23 | 26.20 | 32.80 | 36.80 | 84.67 | 85.00 | 83.67 | 52.24 |
| | MC, $N_{mc} = 1$ | 24.61 | 21.48 | 17.19 | 68.84 | 78.17 | 80.59 | 25.80 | 33.80 | 36.40 | 84.67 | 84.00 | 85.00 | 53.38 |
| LLaDA-Instruct-8B | MC, $N_{mc} = 32$ | 29.69 | 21.88 | 16.41 | 71.11 | 78.70 | 82.79 | 27.60 | **34.80** | 36.20 | 86.33 | 85.67 | 86.67 | 54.82 |
| | **DiSE (ours)** | **30.86** | **24.22** | **27.34** | **73.01** | **82.41** | **83.02** | **29.80** | 34.60 | **38.20** | **88.33** | **87.00** | **90.00** | **57.40** |
| | LLaMA perplexity | 30.86 | 17.19 | 11.33 | 74.22 | 79.61 | 81.20 | 28.60 | 35.40 | 34.80 | 88.33 | 86.67 | 85.67 | 54.49 |
| | Baseline | 24.22 | 15.62 | 17.19 | 70.51 | 77.48 | 79.53 | 26.80 | 34.00 | 36.80 | 87.00 | 84.67 | 86.67 | 53.37 |
| | MC, $N_{mc} = 1$ | 24.22 | 20.31 | 24.22 | 70.13 | 79.45 | 80.89 | 28.20 | 34.80 | 37.60 | **88.33** | 84.67 | 87.33 | 55.01 |
| LLaDA-1.5-8B | MC, $N_{mc} = 32$ | 26.17 | **20.70** | 21.88 | 72.63 | 79.91 | 82.79 | 28.40 | **35.60** | **38.80** | 88.00 | 85.33 | 86.33 | 55.55 |
| | **DiSE (ours)** | **29.30** | 17.97 | **28.91** | **74.53** | **81.96** | **83.55** | **28.60** | 34.40 | 37.40 | 88.00 | **86.33** | **87.67** | **56.55** |
| | LLaMA perplexity | 28.91 | 12.89 | 13.28 | 74.60 | 81.50 | 80.14 | 30.40 | 39.00 | 37.20 | 89.67 | 87.67 | 87.00 | 55.19 |

$P$. For GPQA, we set $M = 1$ for the last seven tokens of the prompt $P$ and the first two tokens of the response $R$. For Countdown, GSM8K, MATH500 and SVAMP, we adopt the selection mode 'last-10' by default, which sets $M = 1$ only for the last ten non-EOT tokens. For flexible-length dLLM generation experiments, we set the maximum number of iterations $M_{max} = 10$, the patience parameter $K = 4$, and the mask size $D = 20$ by default.

# E  ADDITIONAL RESULTS ON UNCERTAINTY QUANTIFICATION

## E.1  BEST-OF-N SAMPLING RESULTS

In Section 5.3, we generate multiple answers for each question and evaluate uncertainty quantification using ROC-AUC scores. As an additional experiment, we perform best-of-N sampling, selecting the answer with the lowest uncertainty (i.e., highest DiSE score in our proposed method) among

Table S2: Additional ROC-AUC results for uncertainty quantification to investigate the impact of different selection modes on performance. We evaluate two selection modes 'full' and 'last-10'. The table reports ROC-AUC scores across the Countdown, GSM8K, MATH500, and SVAMP datasets with varying generation lengths, as well as the average ROC-AUC scores over all settings.

| | | Countdown | | | GSM8K | | | MATH500 | | | SVAMP | | | Avg. ROC-AUC↑ |
|---|---|---|---|---|---|---|---|---|---|---|---|---|---|---|
| | Method / Gen Len | 128 | 256 | 512 | 128 | 256 | 512 | 128 | 256 | 512 | 128 | 256 | 512 | |
| LLaDA-Instruct-8B | DiSE (full) | 0.616 | 0.672 | 0.698 | 0.597 | 0.585 | 0.560 | 0.514 | 0.555 | 0.517 | 0.665 | 0.549 | 0.571 | 0.592 |
| | DiSE (last-10) | 0.578 | 0.521 | 0.622 | 0.633 | 0.644 | 0.658 | 0.611 | 0.634 | 0.604 | 0.688 | 0.692 | 0.755 | 0.637 |
| LLaDA-1.5-8B | DiSE (full) | 0.591 | 0.664 | 0.681 | 0.593 | 0.569 | 0.546 | 0.489 | 0.590 | 0.532 | 0.630 | 0.574 | 0.552 | 0.584 |
| | DiSE (last-10) | 0.610 | 0.471 | 0.586 | 0.610 | 0.616 | 0.613 | 0.606 | 0.553 | 0.533 | 0.599 | 0.629 | 0.677 | 0.592 |

multiple generations per question, and report the accuracy. Consistent with the main experiments, we generate five answers per question. Table S1 presents the evaluation results of our method under the best-of-N sampling strategy on the Countdown, GSM8K, MATH500, and SVAMP datasets with varying generation lengths. The results demonstrate that our approach consistently outperforms the baseline method that does not employ best-of-N sampling across all tested configurations, highlighting the effectiveness of selecting the highest-scoring candidate based on DiSE. In comparison to the conventional Monte Carlo simulation method, our approach yields substantially larger improvements. In particular, when using the LLaDA-Instruct-8B model, the proposed method achieves an average accuracy gain of $5.16\%$ over all twelve generation length settings, whereas the Monte Carlo method with a comparable computational cost, corresponding to $N_{mc} = 1$, achieves only an improvement of $1.14\%$. Even when the Monte Carlo method is applied with $N_{mc} = 32$, resulting in an evaluation cost nearly 32 times higher, the observed improvement reaches only $2.58\%$, which is still considerably lower than the gain provided by our approach. Furthermore, we evaluate performance using probability estimates obtained from an auto-regressive LLM as a reference. For instance, under the same generations, employing the auto-regressive LLM probabilities leads to an improvement of merely $2.25\%$, which remains below the performance enhancement achieved by our method, thereby underscoring the superiority of DiSE in best-of-N sampling and uncertainty quantification. Importantly, the observed improvements are consistent across both tested dLLM variants, LLaDA-Instruct-8B and LLaDA-1.5-8B, across four datasets and three generation lengths. This consistency indicates that best-of-N sampling guided by DiSE remains robust regardless of model, task type or sequence length.

### E.2 ADDITIONAL QUALITATIVE EXAMPLES OF UNCERTAINTY QUANTIFICATION

Figure S1 presents additional qualitative examples of uncertainty quantification using LLaDA-Instruct-8B. Consistently, DiSE effectively distinguishes between correct and incorrect outputs by assigning higher scores to correct answers, corresponding to lower uncertainty, while the Monte Carlo simulation with $N_{mc} = 32$ fails to align with the correctness of the answers. These results provide additional evidence of the effectiveness of DiSE as a fine-grained uncertainty measure at the sequence level.

### E.3 ABLATION STUDY FOR DIFFERENT SELECTION MODES

In Section 5.3, we report the effectiveness of DiSE for uncertainty quantification under the selection mode 'last-10', showing substantial improvements over the baseline across multiple datasets and generation lengths. To further validate the robustness of this finding, we extend the analysis by additionally evaluating the selection mode 'full' configuration and directly comparing it with the mode 'last-10'. The ROC-AUC results are presented in Table S2 and the best-of-N sampling results are presented in Table S3. Without specifying a local region for computing regeneration probability, DiSE with 'full' mode still achieves performance far above the baseline, demonstrating the effectiveness of our method.

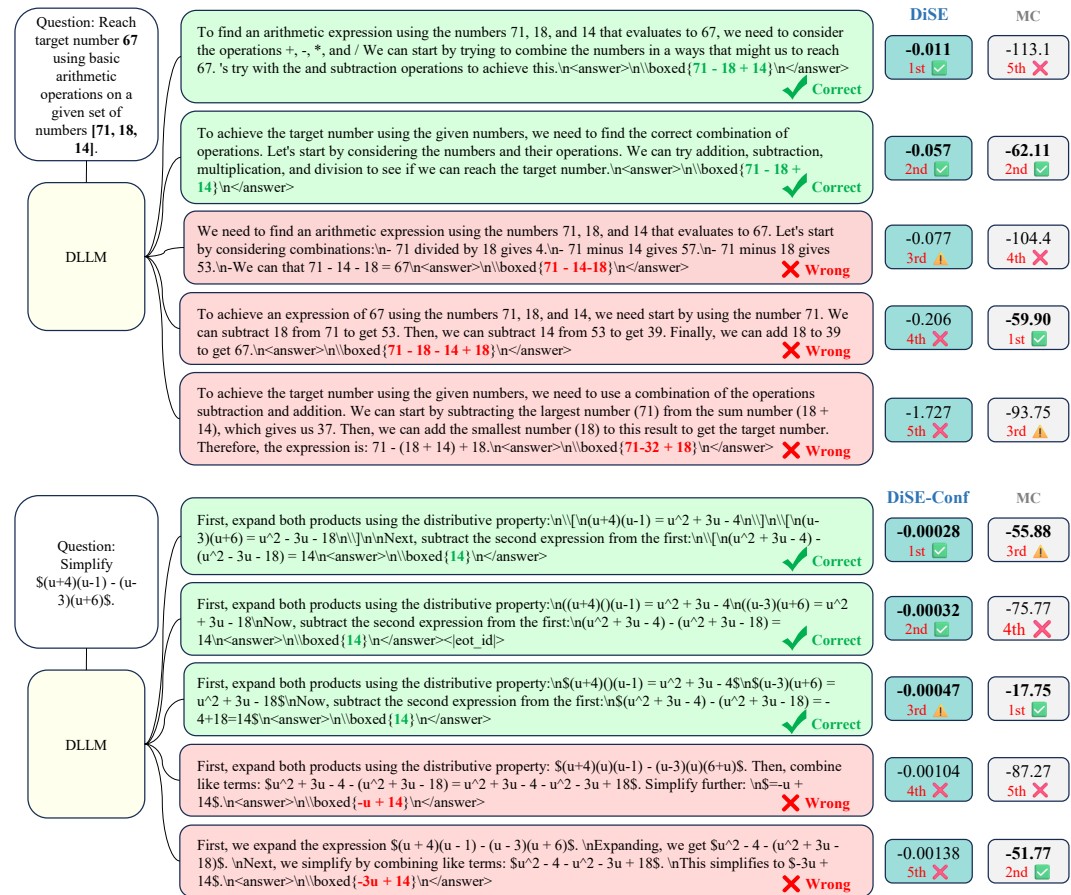

Figure S1: Additional qualitative examples of uncertainty quantification using LLaDA-Instruct-8B. DiSE assigns higher scores to correct answers and lower scores to incorrect answers, while the Monte Carlo simulation ($N_{mc} = 32$) produces scores that do not consistently reflect correctness.

Table S3: Additional best-of-N sampling results for uncertainty quantification to investigate the impact of different selection modes on performance. We evaluate two selection modes 'full' and 'last-10'. The table reports accuracy across the Countdown, GSM8K, MATH500, and SVAMP datasets with varying generation lengths, as well as the average accuracy over all settings.

| | | Countdown | | | GSM8K | | | MATH500 | | | SVAMP | | | Avg. Accuracy |
|---|---|---|---|---|---|---|---|---|---|---|---|---|---|---|
| | Method / Gen Len | 128 | 256 | 512 | 128 | 256 | 512 | 128 | 256 | 512 | 128 | 256 | 512 | |
| **LLaDA-Instruct-8B** | DiSE (full) | 30.86 | 28.52 | 27.34 | 71.87 | 79.76 | 79.53 | 27.20 | 34.60 | 34.20 | 87.67 | 85.33 | 87.00 | 56.16 |
| | DiSE (last-10) | 30.86 | 24.22 | 27.34 | 73.01 | 82.41 | 83.02 | 29.80 | 34.60 | 38.20 | 88.33 | 87.00 | 90.00 | 57.40 |
| **LLaDA-1.5-8B** | DiSE (full) | 27.34 | 25.00 | 32.81 | 72.33 | 79.45 | 80.06 | 24.80 | 37.20 | 38.00 | 88.33 | 86.67 | 85.00 | 56.42 |
| | DiSE (last-10) | 29.30 | 17.97 | 28.91 | 74.53 | 81.96 | 83.55 | 28.60 | 34.40 | 37.40 | 88.00 | 86.33 | 87.67 | 56.55 |

# F ADDITIONAL ABLATION STUDY ON PATIENCE FOR FLEXIBLE-LENGTH dLLM GENERATION

We investigate the effect of different patience values $K$ on flexible-length dLLM generation across the Countdown, GSM8K, MATH500 and SVAMP datasets with varying base lengths, testing under both the LLaDA-Instruct-8B and LLaDA-1.5-8B models. The summarized results are presented in Table S4 and Table S5." Across all tested patience $K$ settings, the flexible-length generation guided by DiSE consistently achieves substantially better average accuracy than fixed-length baselines, demonstrating the effectiveness of adaptive sequence length. Increasing $K$ raises computational

Table S4: Ablation study on flexible-length dLLM generation with the LLaDA-Instruct-8B model, analyzing the impact of different patience values $K$ on performance. The table reports accuracy on the Countdown, GSM8K, MATH500, and SVAMP datasets with varying base lengths, along with the average accuracy and the average number of model forward passes for each configuration.

| Method / Base Len | Countdown | | | GSM8K | | | MATH500 | | | SVAMP | | | Avg. Accuracy | Avg. # Forward Pass |
|---|---|---|---|---|---|---|---|---|---|---|---|---|---|---|
| | 128 | 256 | 512 | 128 | 256 | 512 | 128 | 256 | 512 | 128 | 256 | 512 | | |
| DiSE-flexible (K=2) | 27.34 | 18.36 | 16.02 | 70.43 | 79.30 | 79.23 | 26.40 | 34.00 | 36.60 | 87.33 | 86.00 | 84.67 | 53.81 | 182.3 |
| DiSE-flexible (K=3) | 27.34 | 17.97 | 15.62 | 70.74 | 79.61 | 79.23 | 25.80 | 33.80 | 36.60 | 87.33 | 86.00 | 84.33 | 53.70 | 199.3 |
| DiSE-flexible (K=4) | 27.73 | 18.36 | 15.62 | 70.96 | 79.68 | 79.30 | 26.00 | 33.60 | 36.60 | 87.33 | 86.00 | 84.33 | 53.79 | 215.3 |

Table S5: Ablation study on flexible-length dLLM generation with the LLaDA-1.5-8B model, analyzing the impact of different patience values $K$ on performance. The table reports accuracy on the Countdown, GSM8K, MATH500, and SVAMP datasets with varying base lengths, along with the average accuracy and the average number of model forward passes for each configuration.

| Method / Base Len | Countdown | | | GSM8K | | | MATH500 | | | SVAMP | | | Avg. Accuracy | Avg. # Forward Pass |
|---|---|---|---|---|---|---|---|---|---|---|---|---|---|---|
| | 128 | 256 | 512 | 128 | 256 | 512 | 128 | 256 | 512 | 128 | 256 | 512 | | |
| DiSE-flexible (K=2) | 24.61 | 17.19 | 19.14 | 72.40 | 79.23 | 80.06 | 27.40 | 35.60 | 37.40 | 87.33 | 84.67 | 87.00 | 54.34 | 182.0 |
| DiSE-flexible (K=3) | 24.61 | 19.14 | 20.70 | 72.48 | 79.45 | 80.06 | 27.80 | 35.40 | 37.40 | 87.00 | 84.67 | 87.00 | 54.64 | 198.9 |
| DiSE-flexible (K=4) | 26.17 | 19.53 | 22.27 | 72.33 | 79.53 | 80.06 | 27.20 | 35.60 | 37.40 | 87.00 | 85.00 | 87.00 | 54.92 | 214.7 |

costs, but the corresponding performance gains are not always proportional, highlighting the need to balance efficiency with achievable improvements.

