# OpenReview forum: "Efficient Self-Evaluation for Diffusion Language Models via Sequence Regeneration"
_ICLR.cc/2026/Conference — ICLR 2026 Conference Withdrawn Submission_

### Official Review · Reviewer_1P18 · 2025-10-28

**Soundness:** 2
**Presentation:** 3
**Contribution:** 2
**Rating:** 4
**Confidence:** 4

**Summary:**

process makes quality assessment difficult. DiSE quantifies the model's confidence by calculating the probability of regenerating the tokens in its own output sequence given the full context; a higher regeneration probability signifies greater confidence in the output's quality. Building on this core idea, the research not only utilizes DiSE as an efficient tool for conditional likelihood estimation and uncertainty quantification but also proposes a flexible-length generation framework. This framework leverages DiSE as a real-time self-evaluation signal, enabling dLLMs to dynamically and adaptively determine the optimal output length, thereby overcoming the traditional limitation of fixed-length text generation. Experimental results demonstrate that DiSE significantly improves performance across multiple datasets, increasing likelihood evaluation by 4.0% and uncertainty evaluation by 6.4%, while achieving up to a 32x speedup compared to the conventional Monte Carlo simulation baseline and enhancing the accuracy of flexible-length generation. Ultimately, DiSE introduces an efficient and reliable self-evaluation mechanism for diffusion-based models.

**Strengths:**

1. The method is simple and easy to use.

2. The writing is clear and easy to follow.

**Weaknesses:**

1. The author should obtain dllm in other training methods, such as the effectiveness of DiSE in dream.

2. The author should explore the reasons why DiSE is feasible, rather than simply discovering this phenomenon. From the perspective of llada training, only the prediction of mask tokens will be supervised, while the logits generated by other known tokens are, intuitively speaking, invalid. If the author analyzes this phenomenon, the paper will be more convincing.

3. The author should show the throughput (i.e., generation speed) of flexible generation using DiSE.

**Questions:**

Q1-2: See weakness 1&3

---

### Official Review · Reviewer_6RsG · 2025-10-29

**Soundness:** 2
**Presentation:** 2
**Contribution:** 1
**Rating:** 2
**Confidence:** 4

**Summary:**

The paper proposes the DiSE score, as an alternative to Monte-Carlo (MC) estimation, to evaluate the generation of dLLMs. The authors show that the DiSE score can, in a single forward pass, compare continuations (e.g in MCQ benchmarks), which is more efficient than MC integration. The authors argue that DiSE "facilitates the likelihood computation" (abstract), compared to MC estimation, however do not show how DiSE relates to the true data likelihood.

**Strengths:**

1. DiSE is faster than MC evaluation on the likelihood, and can be used to compare answers to MCQ benchmarks, in a single forward pass, while MC evaluations require many iterations to approximate the true likelihood, and pick the most likely answer.
2. DiSE leads to higher accuracy on GPQA and Math benchmarks, and improves the RoC AUC, compared to MC integration *with few samples (1 or 32)*.

**Weaknesses:**

### Summary of the weaknesses
1. **Likelihood**: DiSE is not shown to estimate or bound the true data likelihood; the reported "gains" in likelihood are not clear vs MC bounds and AR perplexity.
2. **Factual inaccuracy on generation length**: claims that dLLMs require fixed lengths ignore semi‑autoregressive/variable‑length approaches explored in Llada, Plaid, MDLM.
3. **Insufficient citation/positioning**: closely related masked‑LM pseudo‑likelihood work [4-6] is not cited. The existence of these works negatively reflect on the novelty of this work.


### Major weaknesses
1. **Likelihood computation**. In the abstract, the authors argue the DiSE *"facilitates the likelihood computation"*. However, it is not clear how DiSE relates to the true likelihood, or whether it even bounds the log-likelihood, unlike the MC bound present in previous work (MDLM, MD4, RADD, SEDD). Furthermore, still in the abstract, the authors argue the DiSE improves the likelihood evaluation by 4%. What does that mean? Does that mean the likelihood bound is 4% more tight? If so, you need to show that DiSE is a valid bound on the data likelihood.

2. **Factual inaccuracies** (introduction and line 256). While certain dLLMs are trained on fixed-length sequences, prior work has studied flexible-length generation. For example, Llada [1] (which is used in the submission), trains on sequences of varying length during the SFT phase, to handle flexible-length generation. Plaid [2] uses a stochastic length during training which allows the model to generate shorter sequences. Finally, MDLM [3] samples semi-autoregressively, generating text block by block. *These prior work are not sufficiently discussed*.

3. **Missing prior work**: [4-6] have investigated similar ideas, using BERT-style models, which are similar, if not equivalent to masked dLLMs, and trained with cross-entropy. These prior works concluded that BERT-style models can compute a pseudo-likelihood that might capture sentence fluency better than autoregressive scores. These prior works diminish the novelty of the current work.



### Other weaknesses
1. **Conflating likelihood and quality**. Abstract: *"This method enables more efficient and reliable quality assessment by leveraging token regeneration probabilities, facilitating both likelihood estimation and robust uncertainty quantification."* The fact that the model is confident in its generation does not mean that the generation is high quality. For example, GPT-2 will assign a high likelihood, to a repetitive sequence such as "the the the the ...", as it is easy to predict, while it is *not* high quality.
2. **Evaluation details** (lines 199-200): *"We sample 15 well-formed sentences"*. What is a well-formed sentence? Are these extracted from a specific benchmark? Did you write them yourself?
3. **Choice of visualization**: In Figure 2, visualizing the difference of DiSE score between 15 "well-formed" and random sequences, as a 1D sequence of green blocks (the color representing the difference of DiSE score), is not appropriate. Consider using an histogram instead.
4. **Patience $K$**. On line 471, in the experiment section, and shortly before the conclusion, the authors introduce the "Patience" parameter, but do not elaborate on what it represents. This needs to be introduced clearly in the methods section.
5. **Limitations are not discussed**: The authors do not discuss the limitations of their work. While I understand that authors may worry that detailing limitations could be used by reviewers as grounds for rejection, I believe it is important to include some limitations.


[1] Large Language Diffusion Models, Nie et al, 2025

[2] Likelihood-Based Diffusion Language Models, Gularjani et al, 2023.

[3] Simple and Effective Masked Diffusion Language Models, Sahoo et al, 2024.

[4] Pre-Training Transformers as Energy-Based Cloze Models, Clark et al, 2020.

[5] Pseudolikelihood Reranking with Masked Language Models, Salazar et al, 2019.

[6] Masked Language Model Scoring, Salazar et al, 2020.

**Questions:**

1. DiSE relies on model predictions at clean (unmasked) positions, where no loss is applied during training. This may be problematic, as these outputs are unconstrained. Can you clarify how these predictions behave in practice? Are they typically peaked on the true token, or do they have higher entropy compared to masked positions? Some quantitative evidence would be helpful.

2. From line 304, it seems your method generates only one token per forward pass. If so, what is the motivation for using a diffusion LLM instead of an autoregressive model, given that dLLMs cannot benefit from KV caching? Additionally, if the method only generates one token at a time (by masking the last $D$ tokens and generating a single new token), how does this compare in practice to autoregressive generation (e.g., with Llada) in terms of speed and quality?

3. Lines 299-300: *"Let $\bar R$ be the sequence after removing all EOT (did you mean EOS?) tokens from $R$."* How exactly are EOS tokens removed? If they appear at the end, do you truncate the sequence? If they occur in the middle, do you delete or mask them?

4. You stop generation based on the DiSE score instead of the EOS token. Did you compare with a semi-autoregressive baseline that stops at the first EOS? How do performance and speed compare?

5. Lines 377-409: Please clarify what *"a 5.9% gain compared to the perplexity method of an auto-regressive LLM"* means. 5.9% gain with respect to what metric? Note that your approach does not seem to compute a proper perplexity or a valid likelihood bound, unlike MC integration, which is a true bound by variational arguments.

---

### Official Review · Reviewer_Ya5E · 2025-10-30

**Soundness:** 2
**Presentation:** 3
**Contribution:** 2
**Rating:** 4
**Confidence:** 4

**Summary:**

This paper proposes DiSE, a simple and highly efficient method for dLLMs to quantify their own confidence. The core idea is to feed the model's entire output sequence back into itself and calculate the probability of regenerating the existing tokens given the full sequence as context. This regeneration probability serves as a direct measure of the model's confidence in its own output. Meanwhile, the paper introduces a novel flexible length generation framework that uses DiSE to adaptively decide when to stop the generation process. Experiments demonstrate that DiSE increases likelihood evaluation accuracy by 4.0% and uncertainty evaluation by 6.4% on average.

**Strengths:**

1. This paper proposes an effective self-evaluation method DiSE for dLLM that leverages regeneration probability. Compared to the iterative Monte Carlo baseline, which requires numerous forward passes, DiSE only needs a single forward pass.

2. The paper introduces a flexible-length generation framework built on DiSE, which directly addresses the fixed-length generation constraint that typically limits dLLMs.

3. Experimental results demonstrate the effectiveness of the proposed method. DiSE provides more accurate estimations of conditional likelihood and uncertainty, while the flexible length generation framework improves upon fixed-length generation across multiple datasets.

**Weaknesses:**

1. The paper lacks experiments on a broader set of open-source dLLMs (e.g., Dream [1]) to sufficiently demonstrate the effectiveness and generalizability of the proposed DiSE.
2. The experimental details regarding conditional likelihood estimation needs to be further clarified. For example, it is unclear whether the response used in the likelihood estimation is the model-generated output or the ground-truth answer.
3. The experiments on flexible length generation lack comparison with other methods that also support dynamic length generation (e.g., DreamOn [2] and EditFlow [3]). Including these baselines is necessary for a comprehensive performance assessment.
4. The proposed flexible length mechanism appears to have a limitation that it only supports increasing the length and can not support deletion.
5. The performance of the proposed DiSE and flexible length generation is sensitive to hyperparameters. DiSE's performance varies with mode selection for different datasets (Fig. 6). And for flexible length generation, the optimal D is distinct across models (Fig. 7). This will limit generalizability to other dLLMs and tasks.
6. The proposed DiSE needs to "regenerate", i.e., "predict the tokens at positions that are already known." However, the dLLM training loss is calculated only for masked positions [1][4], which implies the model's predictive distribution over the non-masked (i.e., known) tokens is unsupervised. A critical point needs to be addressed: How does DiSE ensure that the "regenerate" probability distribution is reliable? The authors should provide a detailed discussion on this.

[1] Dream 7B: Diffusion Large Language Models

[2] DreamOn: Diffusion Language Models For Code Infilling Beyond Fixed-size Canvas

[3] Edit Flows: Flow Matching with Edit Operations

[4] Large Language Diffusion Models

**Questions:**

See above weaknesses.

---

### Note · Authors · 2025-11-26

I have read and agree with the venue's withdrawal policy on behalf of myself and my co-authors.